# Gene inversion potentiates bacterial evolvability and virulence

Christopher N. Merrikh[1] & Houra Merrikh [1,2]

Most bacterial genes are encoded on the leading strand, co-orienting the movement of the replication machinery with RNA polymerases. This bias reduces the frequency of detrimental head-on collisions between the two machineries. The negative outcomes of these collisions should lead to selection against head-on alleles, maximizing genome co-orientation. Our findings challenge this model. Using the GC skew calculation, we reveal the evolutionary inversion record of all chromosomally encoded genes in multiple divergent bacterial pathogens. Against expectations, we find that a large number of co-oriented genes have inverted to the head-on orientation, presumably increasing the frequency of head-on replication-transcription conflicts. Furthermore, we find that head-on genes, (including key antibiotic resistance and virulence genes) have higher rates of non-synonymous mutations and are more frequently under positive selection (dN/dS > 1). Based on these results, we propose that spontaneous gene inversions can increase the evolvability and pathogenic capacity of bacteria through head-on replication-transcription collisions.

[1] Department of Microbiology, University of Washington, Health Sciences Building J-209, Seattle, WA 98195, USA. [2] Department of Genome Sciences, University of Washington, Seattle 98195 WA, USA. Correspondence and requests for materials should be addressed to H.M. (email: merrikh@uw.edu)

DNA replication and transcription occur simultaneously in bacteria, leading to collisions between the replisome and RNA polymerases (RNAPs)[1–4]. These conflicts collapse the replication fork, break DNA strands, and increase mutagenesis[5–10]. As such, conflicts can significantly influence cellular fitness, replication speed, and genome integrity. To contend with this universal problem, cells have developed multiple essential conflict resolution mechanisms[11–13]. Yet recent work has shown that, even in the presence of fully functional conflict mitigation systems, transcription still collapses the replication fork multiple times per cell cycle[9].

We and others have shown that replication–transcription conflicts can occur in either of the two orientations, leading to different outcomes. Co-directional (CD) encounters occur within genes encoded on the leading strand when replication forks overtake RNAPs[1,14]. The consequences of these encounters are far less severe than head-on (HO) conflicts, which occur in genes encoded on the lagging strand[3,5,11,15,16]. HO conflicts dramatically sensitize cells to even partial depletion of conflict mitigation proteins and increase local mutation rates at the conflict site[5,11,13,17]. In keeping with the severity of these effects, all known bacterial genomes display an orientation bias in which the majority of genes are co-oriented with replication[15,18–20]. Among highly transcribed genes, the case is more exaggerated. In particular, rRNA operons are co-directionally oriented in all known bacterial species[1,15]. These findings indicate that the pressure to minimize HO replication–transcription collisions drives genome organization. Yet, after billions of years of evolution, every known bacterial genome still encodes a large number of HO genes.

The presence of HO genes raises important questions about the evolutionary history of genome architectures and the impact of replication–transcription conflicts. Generally speaking, three patterns could have led to the existing co-orientation bias of each species: (1) A reduction in the number of HO genes from a more balanced ancestral genome, (2) An increase in the number of HO genes relative to an ancestral genome that was predominantly composed of CD genes, or (3) Preservation of the ancestral ratio of HO to CD genes. Our current understanding of conflicts predicts that the first model is more likely to be accurate because negative selection should generally reduce the abundance of HO alleles (and subsequently HO conflicts) over time. However, these models have not been tested.

A reduction in HO genes could occur through gene deletion or inversion to the CD orientation. Between these two possibilities, only inversion events are theoretically capable of preventing HO conflicts without resulting in the loss of important genes. As such, gene inversion events are an optimal means by which the cellular conflict burden could be reduced over evolutionary time. At least two methods for identifying inverted genes/fragments are available: fully assembled genome comparison between strains of the same species, and GC skew analysis[21–24]. Though whole-genome comparison is highly accurate, it is low throughput and requires the computationally intensive comparison of specific genomes[25–27]. Additionally, this method would only identify recent inversion events that occurred since the time the isolates in question diverged. Conversely, GC skew analysis is more efficient, as it does not require genome comparison. Furthermore, this method identifies inversions that occurred during the long-term natural evolution of a given strain, making it a generally superior analysis method.

The technical aspects of GC skew-based gene inversion detection are relatively straight forward. It is well established that guanine is more abundant than cytosine along the leading strand of each chromosome arm, resulting in a positive average GC skew (GC skew = $(G - C)/(G + C)$)[23,28–30]. As this pattern arises due to the mutational footprint of normal DNA replication, the GC skew of any given DNA region reveals its long-term orientation with respect to the movement of the replication fork[21,31]. As such, local inversions in the GC skew are a direct indication that a physical inversion has occurred[21,29]. Though some sequences may naturally defy the average GC skew pattern over short lengths due to functional constraints, this pattern is highly robust over extended lengths. Furthermore, the published GC skew profiles of a variety of bacterial species show numerous local GC skew inversions, suggesting that GC skew analysis can reveal a large number of natural inverted regions[32–34]. Therefore, gene inversion models are testable via the GC skew method.

Here we calculated the GC skew of all chromosomally encoded genes in multiple important clinical pathogens to reveal the global gene inversion record for each species. Our analysis shows that, contrary to expectations, bacterial genomes are consistently gaining new HO genes and operons via inversion. This likely increases the frequency of HO replication–transcription conflicts, which as we previously showed, increase mutation rates within the coding region of Bacillus subtilis genes[6,17]. Consistent with our findings in B. subtilis, we now show that HO genes have a consistently elevated rate of retained non-synonymous mutations in species across phyla. Furthermore, we find that positive selection acts on a higher percentage of HO genes, suggesting that the HO orientation can be beneficial. Based on these data, we propose that most, if not all, bacteria increase the mutability of specific genes through the retention of new HO alleles after spontaneous inversion events. In particular, we find that virulence and antibiotic resistance genes are enriched in the HO orientation. As such, our findings suggest that pathogens commonly harness the mutagenic capacity of HO conflicts, accelerating the evolution of virulence and antibiotic resistance.

## Results

**Physical DNA inversions result in an inverse GC skew**. As previously shown in a variety of bacterial species, guanine nucleotides outnumber cytosine nucleotides within the leading strand of each arm of the chromosome. Using a representative leading strand example sequence, we demonstrate that the GC skew calculation typically results in a positive value (Fig. 1a). Additionally, when this sequence is physically inverted, the GC skew value of the same region concomitantly inverts (Fig. 1a). In Fig. 1b, we show the GC skew profile of the Mycobacterium tuberculosis H37Rv chromosome (green/purple) as a representative example of a bacterial genome. This profile shows multiple GC skew inversion points along each arm of the chromosome, indicating that numerous local fragment inversions have occurred over the course of natural evolution (Fig. 1b, gray arrows point to two examples). To further assess the validity of this method of inversion detection, we computationally aligned 55 fully assembled M. tuberculosis (Mtb) genomes to identify genes that have inverted their orientation relative to the H37Rv reference strain. We then compared the GC skew values of HO and CD alleles of the same genes. Among many examples, we selected four genes that have inverted in specific Mtb lineages (Fig. 1c, Supplementary Data 1). Though the reference strain also possesses inverted alleles, for simplicity we have selected examples in which the ancestral allele (with a positive GC skew) is present in the reference genome. The strains in which inverted alleles were identified are indicated in the figure legend. These data demonstrate that both HO and CD genes are capable of gaining a positive GC skew value (i.e., that negative GC skew values are not caused by gene orientation) like the rest of the genome, in keeping with previous findings[35]. They also show that the ancestral allele/gene orientation can be identified using the GC skew method.

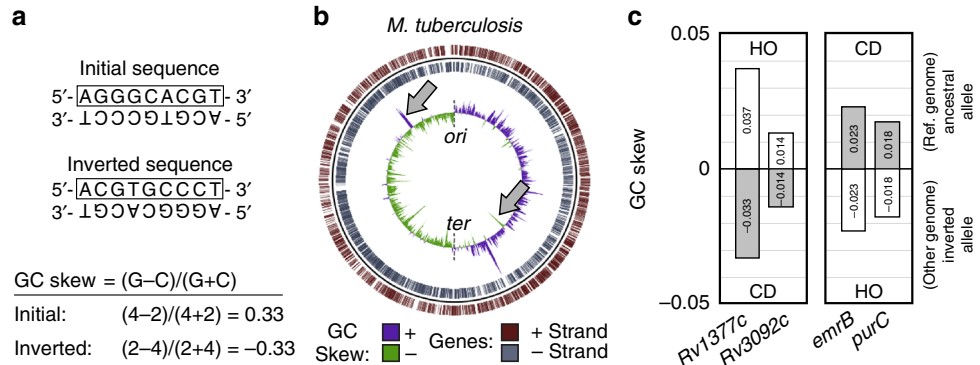

**Fig. 1** DNA fragment inversion concomitantly inverts GC skew values. **a** The effect of DNA fragment inversion on GC Skew. An example DNA sequence oriented with the leading strand on top and its post-inversion form are depicted. The box indicates that the upper strand is used for GC skew calculation by convention. The GC skew calculation shows that the relative abundance of guanine and cytosine residues in the top strand are reversed upon DNA inversion, resulting in an inverse GC skew value. **b** GC skew values along the top strand are plotted on the *M. tuberculosis* H37Rv genome. GC skew values switch from positive (green) to negative (purple) at *ori/ter* regions due to opposing replication fork orientations. Inversions are apparent as deviations from strand averages (two examples are indicated by gray arrows). **c** Anecdotal examples of four genes that naturally inverted in specific *Mtb* strains as identified by closed genome analysis. The ancestral orientation of each allele is indicated above the corresponding column (HO or CD) as inferred from the positive GC skew value. Columns are marked with GC skew values. Gene names are indicated under each column. Inverted alleles (always with both opposing orientations and GC skew values) were found in the following strains: Gene *Rv1377c*: Strain KIT87190, Gene *Rv3092c*: Strain F1, Genes *emrB* and *purC*: Strain 37004

**HO genes frequently have negative GC skew values**. As an initial assessment of DNA inversion patterns, we calculated the chromosomal GC skew of multiple divergent bacterial species using a 100 bp step length (see Supplementary Table 1 for a complete list of species and abbreviations). We then conducted a preliminary visual investigation of various chromosomal regions in which negative GC values predominate (Fig. 2, gray). We immediately noticed that regions with negative GC values tend to correlate with the presence of HO genes or operons. This suggests that many HO genes and operons arose through the inversion of a co-oriented allele. To better assess this possibility, we recalculated the average GC skew of whole-gene regions rather than arbitrary step sizes (Fig. 2, black boxes). This reduces noise that can arise over short step sizes. These data support our initial impression that HO genes frequently have negative GC skew values, whereas most (though not all) co-oriented genes have positive GC skew values.

**Most HO genes were originally co-directional**. To further probe the correlation between the HO orientation and negative GC skew values, we used a genome-wide quantitative analysis. Here we binned GC skew data for each gene by orientation and again by value: either >0 or ≤0 (Fig. 3). This second cutoff reveals the proportion of either CD or HO genes that have been retained in their current orientation (positive values) vs. those that are the product of an inversion event (negative values). For all species, the data are consistent with our initial qualitative observations: Many HO genes have a negative GC skew, which is indicative of inversion from the opposite orientation. In most of the species analyzed, this negative GC skew pattern applied to the majority of HO genes. Conversely, few CD genes have a negative GC skew. These data demonstrate that a significant percentage of HO genes are the product of inversion. Therefore, the data directly oppose the expectation that the production of HO genes should be rare due to their ability to cause harmful HO replication–transcription conflicts.

**HO genes have a higher non-synonymous mutation rate**. Our observation of the wide-spread creation of new HO genes is somewhat counterintuitive as these new alleles should cause HO

replication–transcription conflicts and increased replication stress, resulting in negative selection pressure. The abundance of new HO genes suggests that these expectations are not necessarily correct. Potential explanations for these contradictory findings include the possibility that the new HO genes we identified are not expressed during replication, thereby avoiding conflicts. Conversely, it is possible that the HO orientation confers some benefit outweighing the detrimental effects of conflicts. As HO genes are a diverse group, both explanations may be correct for a subset of genes. Evidence supporting the beneficial hypothesis includes our previous data showing that HO conflicts increase gene-specific mutation rates, and that HO genes have evolved at an accelerated rate during the natural evolution of *B. subtilis*[17]. As such, we previously proposed that the increased mutation rate could provide a net benefit to cells through the more rapid creation of advantageous mutations[17]. This could potentially overcome the negative selection pressure caused by the conflict itself. However, the increased mutation rate of HO genes has only been demonstrated for *B. subtilis*. Therefore, we asked whether HO genes evolved at an accelerated rate in other species. To assess the mutation rates of HO and CD genes in each species, we performed genome-wide mutational analyses in silico using TimeZone software (Fig. 4)[36]. Specifically, we analyzed the rates of non-synonymous (dN) and synonymous (dS) mutations among the core genes of each species, using at least 10 whole genomes per species. Our analysis shows that the rate of non-synonymous mutations is indeed elevated in the HO genes of each species relative to the corresponding CD genes (Fig. 4). Likewise, the average dN/dS ratio is also significantly elevated for HO genes. These data are consistent with our initial analysis in *B. subtilis*, establishing a broadly conserved pattern in which HO genes evolve at an accelerated rate in nature[6,17].

**Positive selection acts on a higher percentage of HO genes**. Though we find that HO genes have a higher mutation rate, mutations are not necessarily beneficial. In other words, the elevated dN and dN/dS ratio are consistent with both increased positive selection which would support the beneficial hypothesis, and decreased negative selection acting on HO genes which would not. To better understand the relative influence of

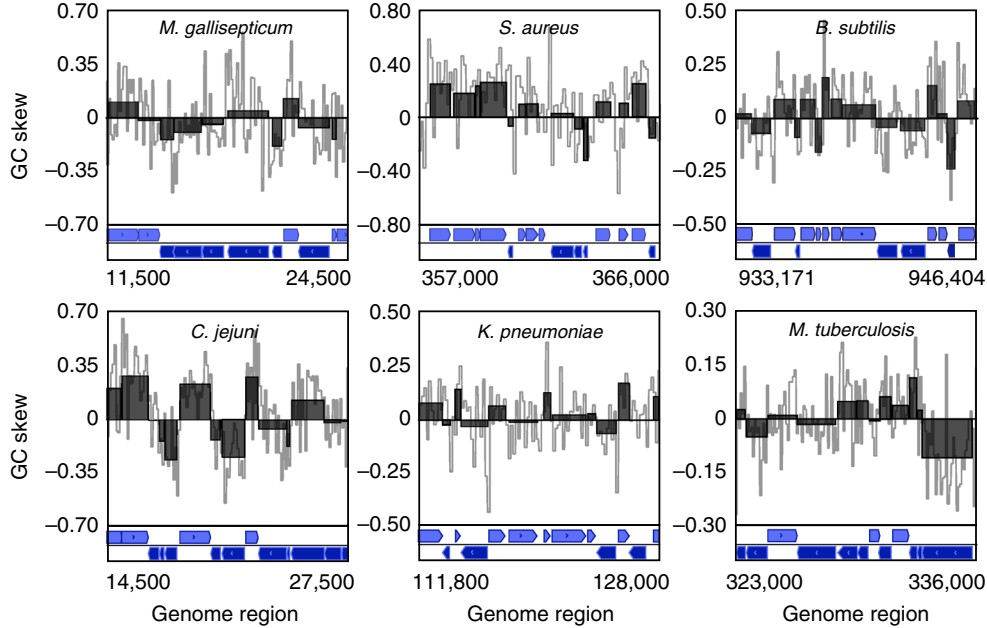

**Fig. 2** Inverted GC skew values correlate with genes in the head-on orientation. GC skew values are either shown in gray (100 bp step size) or as an average over whole genes (black). The locations of individual genes and their coding strand are indicated below. For each species, head-on genes are annotated on the lower box (transcribed right to left, dark purple), and co-directional genes are on the upper box (light purple)

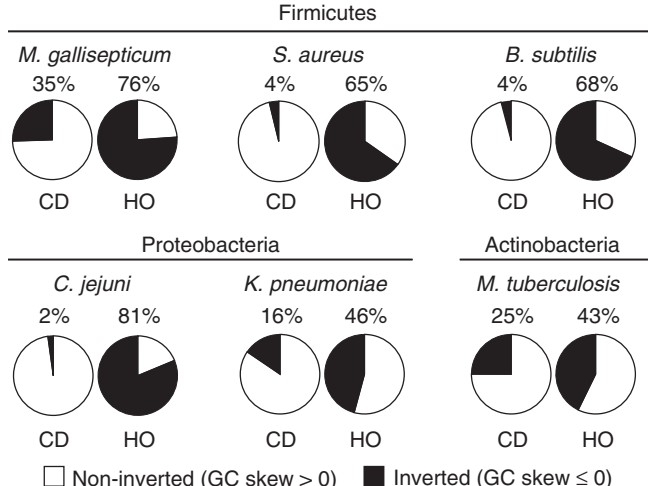

**Fig. 3** A large percentage of head-on genes were originally co-directional. The GC skew values of individual genes were binned by their current orientation in the reference genome and again by GC skew value. Positive GC skew values indicate long-term retention in the current orientation. Negative GC skew values indicate that a gene has inverted from the opposite strand/orientation. GC skew values are also presented in the form of column graphs in Fig. 5. The number of genes in each category is presented in Supplementary Table 2

different selective pressures, we first considered the mutability hypothesis. If HO genes are truly more mutable, a lower percentage of HO genes should meet the conservation cutoff required to be considered core genes (95% conservation in terms of both amino acid identity and gene length). However, we find that a nearly identical percentage of HO and CD genes meet this cutoff in each species (Supplementary Table 2). As such, the data argue against increased mutability of HO genes as an explanation for their higher dN and dN/dS ratios.

Conversely, to identify overt signs of positive selection, we calculated the frequency of HO or CD genes with a dN/dS ratio of >1. This threshold should only be surpassed if a gene is under strong positive selection. We initially observed a higher frequency of genes with dN/dS values exceeding 1 among the HO genes of each species (listed in Supplementary Table 3). Yet the total number of genes in each species was too small to establish statistical significance, necessitating a combined analysis of all data points. This yielded a highly statistically significant difference between the two groups, indicating that positive selection acts on a higher percentage of HO genes (Table 1). Therefore, these data are consistent with positive selection driving the retention of at least some HO alleles after spontaneous inversion from the opposing strand.

Notably, the cutoff value of dN/dS > 1 is a conservative criterion for identifying positive selection. Therefore, we attempted to identify signs that positive selection could be acting on many HO genes (in addition to the 2.4% under overt positive selection). To test this, we manually removed the genes with a dN/dS ratio >1 from our original data set, then re-calculated the average dN/dS values for the remaining genes (Supplementary Table 4). Again, we find that HO genes still have a higher dN/dS ratio, suggesting that it is not only the genes under overt positive selection that are causing the elevated average dN/dS value. These findings are therefore consistent with positive selection acting broadly on genes in the HO orientation.

**HO alleles are retained over evolutionary timescales**. Given the detrimental effects of HO transcription, it is possible that new HO alleles may rapidly revert to the CD orientation. This model would argue against the beneficial HO gene hypothesis we proposed above because it would suggest that HO conflicts are, in essence, avoided. Alternatively, evidence of HO gene retention after inversion would support the beneficial HO orientation model. Previous work shows that the negative GC skew of inverted DNA fragments rises over time due to normal DNA replication, just like any other DNA fragment[21]. Therefore, if the inverted regions identified here are retained in the new

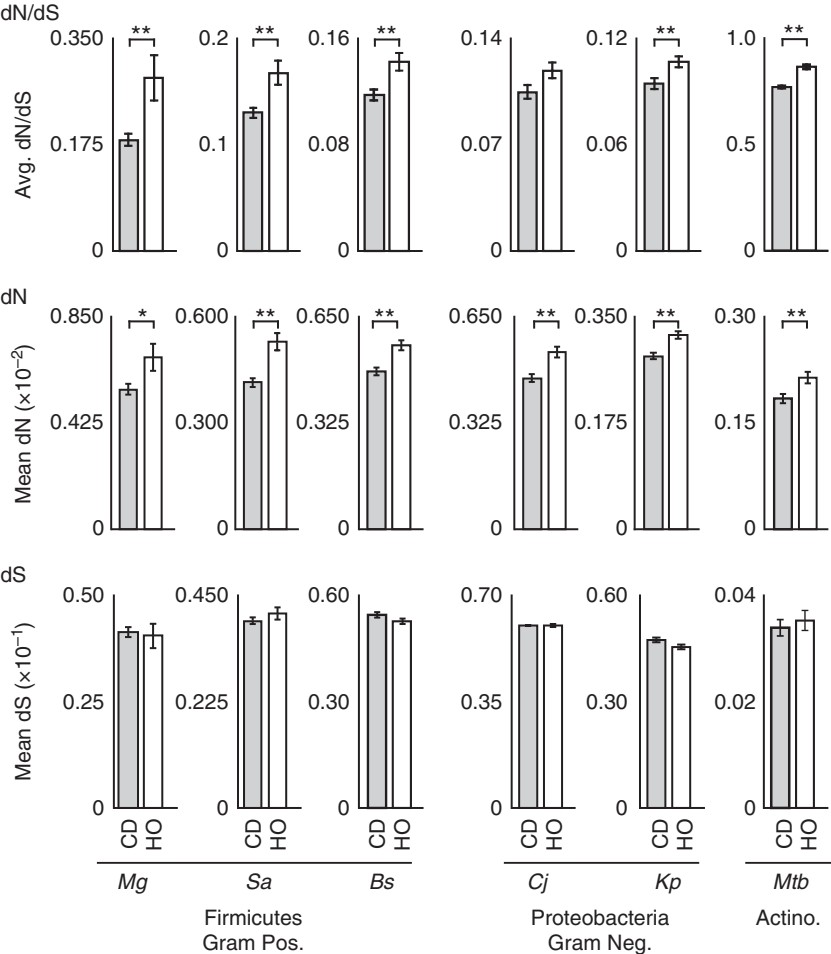

**Fig. 4** Head-on genes evolve at an accelerated rate across phyla. Whole-genome mutational analysis including the dN/dS ratio, rate of non-synonymous mutations (dN), and rate of synonymous mutations (dS) are shown. Values were calculated for core genes that are ≥95% conserved in both length and amino acid sequence using at least 10 complete genomes per species. Additional species information including naming abbreviations and the number of core genes analyzed per species are available in Supplementary Table 2. Error bars represent the standard error of the mean. Statistical significance was determined by the two-tailed Z-test. *p < 0.05. **p < 0.01

orientation, their GC skew should have increased over time. Conversely, if inverted regions rapidly re-invert to the original orientation, their average GC skew value should mirror that of genes in the original orientation. To assess these possibilities, we compared the average GC skew of retained CD genes (those with a positive GC skew) to HO genes that are the result of an inversion (those with a negative GC skew) (Fig. 5). These data show that the two groups do not have equivalent average GC skew values. Specifically, the HO gene average is significantly higher than predicted (the magnitude is decreased) by the rapid inversion hypothesis (i.e., the null hypothesis). As such, these data are consistent with the possibility that at least some HO alleles are retained for significant periods of time following inversion, and that this orientation could be beneficial.

**HO genes are enriched in common functions across species.** To better understand the implications of CD to HO gene inversions, we sought to identify the functions of HO genes. Here we used the DAVID bioinformatics database to identify functions enriched in the two orientations in each species (Fig. 6). To better identify trends (if any) in these functions, we expanded the number of species used for this analysis (listed in the Methods

section). We first noted that a lower number of functional categories are enriched among lagging strand genes: a ratio of approximately 1:3 for HO vs. CD genes, respectively (Fig. 6a). Interestingly, the magnitude of this effect is consistent between low and high co-orientation bias organisms. For example, Gram-positive species possess roughly 1 HO gene for every 5 CD genes, whereas the ratio is as low as 1:1.4 for low co-orientation bias organisms, such as Gram-negative species. The consistency in the relative number of enriched functions across these diverse species suggests that gene function may be an important factor driving the segregation of genes between the leading and lagging strands.

We then found that the same functions are often enriched among HO or CD genes even between highly divergent species (Fig. 6b). Again, this strongly suggests that lagging strand encoding is non-random. Enriched functions include transcription regulation, trans-membrane localization, cell signaling, and virulence. Notably, though not overtly identified via the DAVID database analysis, we also find that a considerable number of antibiotic resistance-related genes, including multidrug resistance pumps, are encoded on the lagging strand (Supplementary Table 5). This suggests that many virulence and antibiotic resistance genes evolve at an accelerated rate due to their HO orientation.

**Table 1 Positive selection acts on a higher percentage of head-on genes**

|    | Total genes | # Genes dN/dS > 1 | Percentage | Chi-sq. p value |
|----|------------|-------------------|------------|------------------|
| CD | 15,627     | 289               | 1.8        | 0.0039           |
| HO | 9757       | 234               | 2.4        |                  |

The number of genes with a dN/dS ratio exceeding 1 were tallied across all six species and combined. A two-tailed Chi-squared test was used to determine significance

**Many virulence and antibiotic resistance genes are HO**. Together, our data set up the expectation that HO conflicts increase the mutation rate and accelerate the evolution of many virulence and antibiotic resistance genes. Consistent with this model, we previously showed that several stress response and virulence genes are HO in *Listeria monocytogenes*[11]. Accordingly, we also found that the conflict resolution factor RNase HIII is essential for efficient survival and replication of bacteria during mouse infections and in vitro infection of activated macrophages[11]. Here we provide additional evidence that a variety of antibiotic resistance and virulence genes anticipated to be expressed during infection are HO (Table S6). Anecdotal examples include: the *Mtb* multidrug resistance pump gene Rv1458c, which confers potent resistance to multiple front line antibiotics when overexpressed (dN/dS = 2.72)[37]; the *Campylobacter jejuni* siderophore transport protein *tonB3*, which is required for infection[38]; the *C. jejuni* distending toxin genes *cdtABC*[39]; the *Staphylococcus aureus* master virulence regulator *traP* and *agrA*, which together regulate the small RNA virulence gene regulator, RNA III[40,41]. These examples serve to demonstrate that critical virulence genes are likely subject to HO conflicts and accelerated evolution.

In light of the possibility that virulence genes may be key targets of HO replication–transcription conflicts, we chose to further probe the effects of orientation on their mutation rates. Though it is formally possible to measure the rate of virulence and antibiotic resistance gene evolution, this can be technically challenging. For example, the virulence and antibiotic resistance genes of many species are acquired as pathogenicity islands or plasmids. As such, not all strains possess the virulence genes that confer pathogenicity. Furthermore, we have excluded plasmids from our analysis as they are often replicated using rolling circle replication, which is not well characterized in the context of replication–transcription conflicts. Therefore, we turned our attention to *Mtb*, which is not thought to acquire plasmids or other foreign DNA. Instead, *Mtb* encodes all of its antibiotic resistance and virulence genes on the chromosome. Here we parsed *Mtb*'s core antibiotic resistance and virulence genes by orientation and calculated the mutation rates of these genes (Tables 2 and 3, respectively). These data clearly show that genes of the same function have a higher mutation rate when oriented HO to replication as shown by the elevated dN/dS ratio and higher frequency of genes under overt positive selection (dN/dS > 1 genes). Therefore, these data are consistent with the model that virulence and antibiotic resistance genes evolve at an accelerated rate through HO conflicts.

## Discussion

Previous studies have demonstrated that most, if not all bacteria, avoid the worst HO conflicts by co-orienting the transcription of highly expressed genes with replication[18,19]. This co-orientation bias should protect the cell from extremely severe constitutive HO conflicts. Yet, the gene inversion patterns identified here indicate that the complete abolishment of HO transcription is not

advantageous. Instead, our findings highlight a surprising and fundamental pattern in the evolutionary history of bacterial species: the widespread creation of new HO genes and operons via inversion. These observations are particularly interesting in light of a large body of work on replication–transcription conflicts, which collectively set up the expectation that new HO alleles should generally experience negative selection pressure due to their tendency to cause severe replication stress[1,2,11,42,43]. The data presented here directly contradict this expectation. As such, our work offers a major fundamental insight into the evolution of genomic architectures. We found that a higher percentage of HO (relative to co-directional) genes are under positive selection. This suggests that the higher mutation rate of HO genes could be beneficial to the cell through the more rapid sampling of new mutations. However, it is important to re-iterate that we are not proposing that the HO orientation is "used" to promote adaptation. Instead, we suggest that a random CD to HO inversion event will create two competing populations, each harboring the same gene but in opposing orientations. If the cells harboring the HO allele gain a beneficial mutation first (a significant possibility due to the higher mutation rate), these cells should increase in abundance, amplifying the prevalence of the HO allele. In other words, the retention of HO alleles should be the result of second-order selection, also known as the hitchhiker effect[44,45]. This model also serves to explain how genes under positive selection could become enriched in the HO orientation—cells harboring HO alleles that are not under positive selection should diminish within populations due to negative selection pressure conferred by HO conflicts. Conversely, those with HO alleles under strong positive selection should increase in prevalence (This assumes that the benefit of the HO mutation(s) are sufficient to overcome the negative selection conferred by the conflict-mediated increase in replication stress.).

Evolvability, or evolutionary potential, can be defined as the relative ability of a particular strain to increase its fitness after evolving over a defined time period[44]. Experiments have demonstrated that, in certain circumstances, increased mutation rates can increase evolvability, presumably through the more rapid creation of beneficial mutations (and despite the increased creation of more neutral and detrimental mutations)[44–48]. Based on both the data presented here, and previous work on conflicts, we propose that the continued inversion of genes to the HO orientation may increase the overall mutation rate of a given species. This should have the effect of increasing a given species' evolvability or potential to evolve. To put it differently, existing strains that harbor numerous new HO alleles are likely more evolvable than a hypothetical case in which the same strain retains the original CD alleles.

There are limitations to this model. For example, some genes do not appear to be appropriate for HO encoding, such as some highly conserved genes[18]. Therefore, we are not suggesting that all genes can be inverted to increase evolvability. Instead, we suggest that the inversion sampling that occurs on a population level allows cells to identify inversions that are either neutral (no benefit, but no conflicts) or beneficial. It is also important to note that we are not suggesting that organisms with few HO genes evolve more slowly than species with many (i.e., Gram-positive versus Gram-negative bacteria, respectively) as mutation rates are not, on the whole, higher for Gram-negative species[49,50]. Instead, mutation rates have been shown to be species specific. As such, the mutagenic effect of the HO orientation is only observable when comparing HO and CD genes within the same species.

As this study is particularly concerned with the inversion of whole genes and operons, we considered possible mechanisms through which the inversion of whole genes, rather than gene fragments, might occur. In particular, we looked for homology

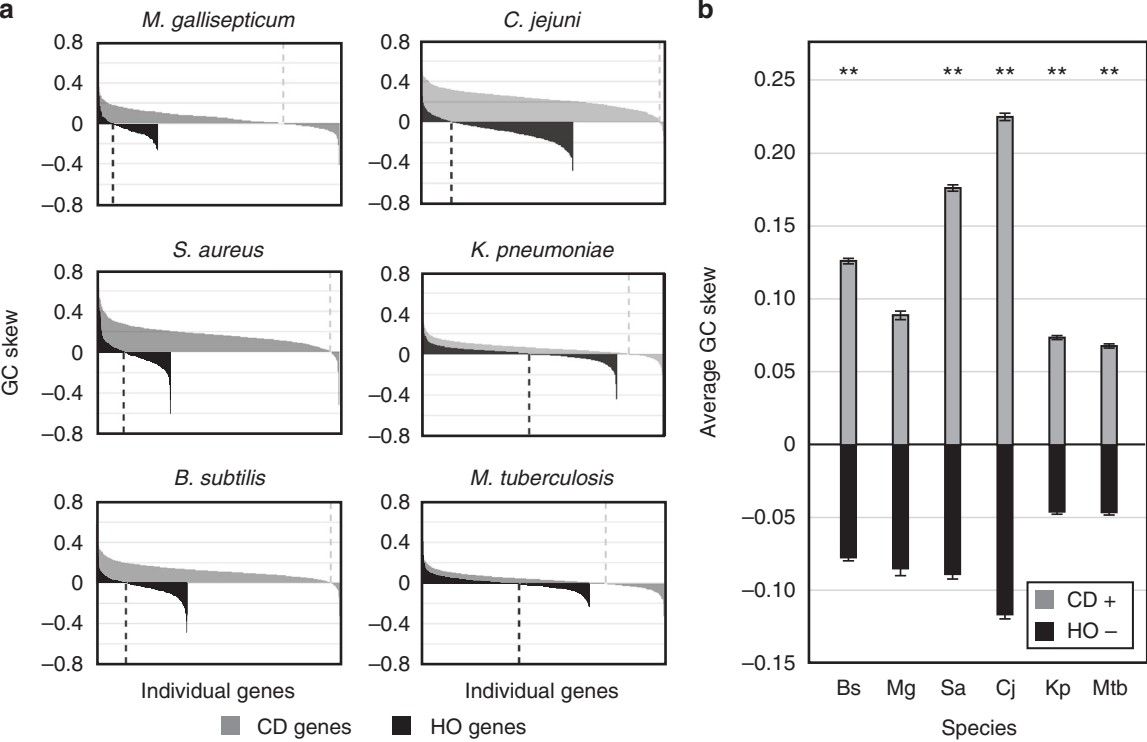

**Fig. 5** New head-on alleles are retained after inversion. **a** The distribution of GC skew values for all co-directional (leading strand, gray) or head-on (lagging strand, black) genes are plotted, one column per gene. Columns are sorted from the highest to lowest GC value resulting in the appearance of a curve. Values below zero represent inverted genes, whereas values above zero suggest long-term orientation conservation. Dashed lines highlight the inflection point between positive and negative value in each group. **b** The magnitude of the average GC skew of inverted head-on genes (those with a negative GC skew, black) is diminished relative to retained co-directional genes (those with a positive GC skew, gray). Error bars represent the standard error of the mean. Statistical significance of the difference in magnitude between the two groups (CD+ vs. HO−) was determined per species using Z-test. **p < 0.0001

regions upstream and downstream of gene regions that could be used for recombination. We noticed that promoter and 3′ UTR regions, by virtue of their AT richness, should have lower sequence complexity than regions in which all four nucleotides are equally distributed, i.e., within open reading frames. The depletion of guanine and cytosine in these regions should therefore increase the likelihood that upstream and downstream regions will be homologous. As such, the AT richness of promoters and 3′ UTRs should be protective to open reading frames by promoting whole gene or operon inversion by homologous recombination[51].

Previous investigations of gene orientation have concluded that essential and highly expressed genes are preferentially co-directional[18,19]. However, our observation that there is an an equal percentage of highly conserved core genes in both the CD and HO groups suggests that essentiality does not drive genes to become co-directional. This contradiction between our work and previous findings is likely because the previous study erroneously identified many HO genes as non-essential[52]. The previous study considered a gene essential only if a deletion strain is not viable under laboratory conditions. However, we and others find that core gene analysis (conservation) is a more informative measure of essentiality[53]. For example, the HO B. subtilis rnhC gene that encodes RNase HIII is considered non-essential. Yet, under mild stress conditions (e.g., high salt that is not lethal in wild-type cells), RNase HIII is absolutely essential[11]. Similarly, in L. monocytogenes RNase HIII (again HO) is non-essential during growth in rich media, yet it is essential during pathogenesis[11]. Based on these data and our functional enrichment analysis, we

conclude that in nature, gene function, rather than essentiality, drives the segregation of genes into the two orientations.

There are multiple possible explanations for the enrichment of specific types of genes in the HO orientation. These include greater mutability (i.e., they are better suited to handle the higher mutation rate), transcriptional repression during replication (i.e., HO conflicts are not produced), and access to a higher number of mutations capable of providing a benefit to the cell. As HO genes encompass a diverse group, all three possibilities may be correct for a different subset of genes. However, the first two hypotheses describe a situation in which the HO orientation is neither harmful nor beneficial to a particular group of genes. Under these hypothetical situations, genes should be evenly distributed between the two orientations, rather than enriched in one. As such, only the beneficial hypothesis seems sufficient to explain the enrichment of specific types of genes in the HO orientation.

The consistent gene functions enriched in the HO orientation across highly divergent species raises the possibility that we are observing convergent evolution. In other words, different species may have independently arranged a subset of virulence and antibiotic resistance genes, among others, in the HO orientation. As convergence is one of the strongest indicators of adaptive evolution, future identification of convergent inversion events would lend strong support to the adaptive hypothesis. However, it is also possible that divergent species simply retained the HO genes provided by a common ancestor. As such, future phylogenic studies are needed to distinguish between these possibilities.

Finally, our discovery that virulence and antibiotic resistance genes are enriched in the HO orientation is consistent with the

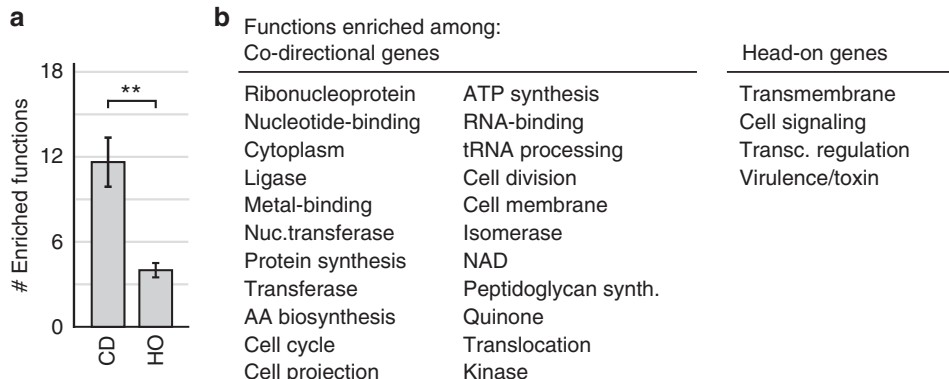

**Fig. 6** Across phyla, HO genes are enriched in fewer and consistent functions. **a** The relative number of functions identified as enriched among co-directional (CD) vs. head-on (HO) genes is highly consistent across phyla. The number of functions found to be enriched among CD or HO genes was determined for 12 species (listed in the Methods section), then averaged across species. Error bars represent the standard error of the mean. Student's $t$ test was used calculate statistical significance, **$p$ value = 0.0004. **b** The functions of all lagging strand (head-on) or leading strand (co-directional) genes were determined on a per species basis. Functions found to be enriched in ≥4 organisms were included here as consensus functions

### Table 2 HO drug resistance genes evolve at an accelerated rate

| | *Mtb* drug targets | |
| | Avg. dN/dS | % dN/dS >1 |
|---|---|---|
| CD | 0.81 ± 0.16 | 20% |
| HO | 1.21 ± 0.21 | 50% |

The mutation rates of core gene drug targets in multi, extensively, or totally drug-resistant Mtb strains were analyzed by gene orientation. Genes under overt positive selection are indicated as the fraction of genes with a dN/dS value >1. Drug targets were previously compiled by Hameed et al.[57]. Among these, we analyzed the 5 head-on and 10 co-directional core genes. Significance was determined by Student's $t$ test. $p$ = 0.2 for HO vs. CD average dN/dS values

### Table 3 HO virulence genes evolve at an accelerated rate

| | *Mtb* virulence genes | |
| | Avg. dN/dS | % dN/dS >1 |
|---|---|---|
| CD | 0.77 ± 0.20 | 33% |
| HO | 1.52 ± 0.27 | 58% |

Mutation rates of virulence genes were analyzed by gene orientation (15 co-directional and 12 head-on core genes). Student's $t$ test $p$ = 0.035 for HO vs. CD average dN/dS values

concept of the "everlasting" host–pathogen arms race[54,55]. This colloquial description refers to the constant requirement for pathogens to rapidly create adaptive mutations in order to evade changing host immune responses. These principles are further supported by the observation that virulence and antibiotic resistance genes that are HO have evolved at an accelerated rate and are more frequently under positive selection than their co-oriented counterparts, at least in *Mtb*. This also implies that proteins encoded by co-oriented genes may be superior drug targets due to the lower mutation rate and presumably decreased ability to gain resistance-conferring mutations. Conversely, researchers should pay particular attention to HO virulence genes as they may represent key sources of genotypic and phenotypic diversity during infection.

In summary, based on the many forms of evidence presented here, we conclude that leading to lagging strand gene inversion (CD to HO) is a common phenomenon during the natural evolution of bacterial species. New HO alleles may cause additional HO replication–transcription conflicts, a higher gene-specific mutation rate, and accelerate evolution. In particular, many genes encoding transcription regulators, trans-membrane proteins, virulence factors, and antibiotic resistance proteins appear to be well suited to HO encoding. As such, gene inversion events appear capable of driving increased bacterial virulence and drug resistance in a variety of clinically important bacterial pathogens.

## Methods

**Chromosome mapping**. gView version 1.7 was used to map the GC skew and genes in Fig. 1[56].

**Ortholog comparison**. TimeZone v1.0 software was used to identify orthologs between 55 fully assembled *Mtb* genomes[36]. Custom Python scripts were used to mine strand and location information for each ortholog from corresponding Genbank files to identify inversions.

**Mutational analysis**. Ten or more fully assembled genomes were analyzed per species using TimeZone v1.0[36]. Core genes were defined as having 95% similarity in amino acid content and gene length. Custom Python and Matlab scripts were used to analyze the location of *ori* and *ter* (inflection points in the chromosomal GC skew map) to assign gene orientation. Analyzed genomes are listed in Supplementary Table 6.

**Functional enrichment analysis**. The DAVID bioinformatics database was used to analyze functions enriched among all HO or CD genes (not just core genes) of 12 species: *Mycoplasma gallisepticum* (we use *M. gallisepticum* as a proxy for the human pathogen *Mycoplasma genitalium*. Few genomes are available for *M. genitalium*, precluding a robust analysis), *B. subtilis*, *S. aureus*, *L. monocytogenes*, *Clostridiodes difficile* (formerly *Clostridium difficile*), *Pseudomonas aeruginosa*, *Klebsiella pneumoniae*, *Escherichia coli*, *Borrelia burgdorferi*, *C. jejuni*, and *M. tuberculosis*. The output list of functions was manually edited to reduce redundant functions. Specifically, we combined the following: membrane, transmembrane, and membrane transport; transcription and transcription regulation; cell cycle and cell division; amino acid synthesis and the synthesis of any specific amino acid; and metal binding and the binding of any specific metal ion.

**Virulence and antibiotic resistance gene mutation rates**. Mycobrowser (https://mycobrowser.epfl.ch/) was used to collect a list of *Mtb* virulence genes. Specifically, genes under key words "host," "antigen," "invasion," and "virulence" were compiled. Genes contributing to drug resistance in *Mtb* strains were previously compiled by Hameed et al. (2018) (Table 2)[57]. Mutation rate data from core gene analyses using TimeZone were used to calculate gene orientation-specific dN/dS values for genes in each list.

**Code availability**. Custom Python scripts used in these analyses are available upon request. Script names and functions are listed in Supplementary Table 7.

## Data availability
Primary data files are available upon request including GC skew data used in Figs. 3 and 5, TimeZone analysis files used in Fig. 4, and gene functions used in Fig. 6.

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

## Acknowledgements

We would like to thank Sarah Mangiameli for her help with the *ori* and *ter* assignments. This work was supported by the Bill & Melinda Gates Foundation, Grant#OPP1154551, and National Institute of Health Director's New Innovator Award to H.M. (DP2GM110773).

## Author contributions

C.N.M. and H.M. designed experiments and analyses. C.N.M. performed all analyses. C.N.M. and H.M. wrote the manuscript.

## Additional information

**Competing interests:** The authors declare no competing interests.

