## [Peer Review File · Nature Communications]

Reviewers' comments:

Reviewer #1 (Remarks to the Author):

This is an interesting and thought-provoking study addressing the role of transcription-replication collisions in bacterial evolution. The main point is that large number of ancestral co-directional genes have inverted to the head-on orientation. The latter head-on genes have higher dN/dS ratio and are more frequently under positive selection. The authors proposed that bacteria increase their evolution rate through gene inversion resulting in head-on transcription-replication collisions.

The key advance of this study is the calculation of the GC skew for all chromosomally encoded genes in multiple important clinical pathogens. This approach allowed the authors to nicely document the inversion record of every single gene in each species. They found that bacterial genomes are consistently gaining new head-on genes by inversion of co-directional genes. In fact, their data indicate that major percentage of current head-on genes were originally co-directional.

They then analyzed the rates of nonsynonymous (dN) and synonymous (dS) mutations among the core genes of each species, using at least 10 whole genomes for each. It appeared that dN/dS ratio is elevated in the head-on genes of each species relative to the corresponding co-directional genes. Furthermore, the fraction of genes with the dN/dS ratio above 1 (an indicator of the positive selection) is significantly higher in these head-on genes than in their co-directional counterparts. Finally, these head-on genes are enriched in common functions across species; specifically, they include important virulence and antibiotic resistance genes.

Overall, this is a very interesting study: the data are novel and they are by-and-large in agreement with the authors' conclusions. I have, however, few questions/comments that the authors may want to address.

First, DNA segments that undergo inversions are usually flanked by inverted DNA repeats. Do the authors see IR around co-directional genes that tend to undergo inversions?

Second, a thematically close study [Sankar et al (2016) The nature of mutations induced by replication–transcription collisions Nature 535: 178-181] found two mutational signatures for the head-on transcription replication collisions: duplications/deletions at the 3'-part of a head-on reporter and base substitutions in its promoter. Do the authors' analysis reveal these mutational signatures? Whatever the answer, this matter needs to be discussed.

Third, I cannot grasp how the authors' hypothesis can explain dramatic difference in terms of collisions between Gram-positive and Gram-negative bacteria. The authors are well aware of this difference stating that " Gram-positive species possess roughly 1 head-on gene for every 5 co-directional genes, whereas the ratio is as low as 1:1.4 for Gram-negative species". In light of the authors' hypothesis this would mean that Gram-negative bacteria evolve much slower than Gram-positive bacteria, which is hardly the case. I'd like the authors to discuss this paradox.

Reviewer #2 (Remarks to the Author):

In the current manuscript, Christopher Merrikh and Houra Merrikh report a novel link between the chromosomal direction of bacterial genes and the adaptability of their host organisms. It is a well-known fact that the gene composition of the leading and the lagging strand of the bacterial

chromosomal DNA is different. On the lagging strand the direction of the replication and transcription machinery is opposite resulting frequent collision between the replisome and the RNA polymerases. The collision causes the collapse of the replication fork, subsequently leads to DNA break and thereby increases mutation rate in the so-called head-on oriented genes located on the lagging strand. In order to avoid the detrimental effect of machinery collision, the majority of bacterial genes are located on the leading strand; hence their transcription is co-directional with the replication. It is generally assumed that there is a negative selection pressure against gene inversion events which is responsible for the head-on orientation of genes. Accordingly, frequently-expressed and essential genes tend to be co-directional (Wen-Xin Zheng et al, 2015)

Christopher Merrikh and Houra Merrikh aim to challenge the above-mentioned findings in their recent manuscript. The authors report that the head-on orientation of specific genes might be beneficial in certain cases by increasing the likelihood of the accumulation of adaptive mutations. They claim: The head-on orientation of certain genes causes increased mutation rate that helps the adaptation of cells.

The head-on gene orientation is very frequent among virulence-related and antibiotic resistance-conferring genes.

The head-on gene orientation is a previously uncovered mechanism among bacteria that can generally contribute to the rise of antibiotic resistance.

I found the subject of the paper interesting, but the general conclusions are not supported by their results. The presented results failed to convince me that the head-on gene orientation has positive effect and thereby it is adaptive. The results only demonstrate that the head-on gene orientation is not always detrimental.

1. The authors only showed that virulence-related genes are enriched among head-on oriented genes, but they did not demonstrate any bias toward head-on direction within the virulence-related gene group.
2. The results demonstrate that the resistance-conferring genes can tolerate the head-on orientation, but there is no obvious that head-on inversion of originally co-directional genes can be beneficial for pathogenic bacteria.

In general, I find the theoretical argument unconvincing. Do the authors really believe that inversion events are favored by selection for the future benefit of increasing evolvability? In my view it is implausible, because any potential selective advantage would be very low, for two reasons. First, the number of genes with increased mutation rate by this mechanism is limited. Second, it has remained unclear to what extent head-on orientation increases local mutation rate. Even more worrying, that there is a simple alternative possibility that needs to be considered. Head-on orientation is overall disadvantageous (e.g. due to interference with transcription), and only genes with relaxed selection can reside in the lagging strand.

Overall I find that the manuscript in its present form does not reach the high standard of Nature Communication, as the conclusion is not well-supported by the experiments and analyses. My concerns and suggestions are found below.

Major comments:

1. In order to detect gene inversion events, the authors used GC skew calculation. They mention that this method is 'well-established' and 'previous work shows that such inversions can be identified by local changes in the GC skew'. However, they did not describe why they chose this method for their analysis.

a) What are the alternative methods to identify gene inversion events?

b) Why is the GC skew analysis assumed to be the best method to identify gene inversions (if it is)?

c) What are the limitations or the shortcomings of the GC skew analysis?

2. In the Introduction section the authors mention that in many cases positive selection acts on head-

on genes. They showed that head-on genes tend to have a high non-synonymous/synonymous mutation ratio ($dN/dS > 1$). However, high dN/dS ratio around one may also reflect relaxed negative selection. Consistent with this hypothesis, non-essential genes tend to be head-on oriented. Genes under relaxed negative selection are expected to be less sensitive to the elevated mutation rate, therefore the head-on orientation can be preserved during evolution. In the Chapter 'Head-on genes are retained over evolutionary time scales' the authors have demonstrated the same: inverted genes have been in head-on orientation for a long time.

3. The authors refer to literature that 'local DNA inversions can spontaneously occur in all living cells' and they write in Introduction section that 'bacterial genomes are consistently gaining new head-on genes and operons via inversion'. However, the appearance of new head-on oriented genes is not clearly proved by the results and the authors have only quoted 'anecdotal examples' demonstrating 'important virulence and antibiotic resistance genes are head-on' (last chapter of Results section). The authors should demonstrate the ratio of co-direction to head-on conversion generally as well as within virulence-related genes.

A higher ratio of head-on orientation of virulence-related genes in pathogenic bacteria would support the hypothesis that head-on orientation of specific genes is indeed preferable in some cases. Furthermore, one might also expect that the gene inversion of resistance-related genes should be more frequent in bacteria that encountered antibiotic exposure in the clinic than in those that did not. Without the above-mentioned demonstration, I am not convinced that abundant co-directional to head-on inversions due to positive selection pressure have been occurred recently.

4. One of the main findings of the manuscript is that the head-on orientation of genes increases the virulence and antibiotic resistance in bacteria. At the end of Introduction section, the authors write that 'bacteria are universally increasing the mutability of their genome via inversions' and it is a 'commonly harnessed' procedure among bacteria. However, I find that this conclusion is premature and exaggerated. Later on, the authors themselves restrained their previous statement in the Chapter 'Head-on genes are enriched in common functions across species': 'This suggests that many virulence and antibiotic resistance genes may evolve at an accelerated rate, at least partially due to their head-on orientation.' These claims are not directly supported by the data.

5. The authors conclude that special functions are enriched among the head-on genes. They found only four cell-physiology functions which are enriched among the head-on genes including toxin and virulence genes. They claim that bacteria adapt faster by the higher mutation rate of these special genes. In my opinion; the authors should analyze the average mutation rate of virulence-related genes irrespective of their orientation. This analysis would be able to clarify, whether the higher mutation rate of virulence-related genes is a consequence of gene orientation or the gene function itself.

Minor comments:

1. Lack of references.

The authors refer to previous results many times without references in the Result section. It is not clear, whether or not these are the authors' previous studies.

Page 4. 'As previously shown in a variety of bacterial species, guanine nucleotides outnumber cytosine nucleotides on the leading strand of each arm of the chromosome.'

Page 9. '...we previously showed that head-on conflicts increase gene specific mutation rates and accelerate evolution in *B. subtilis* in nature.'

Page 12. 'Previous work demonstrated that the negative GC skew of inverted DNA fragments rises due to normal replication, just like any other DNA fragment.'

Page 6. 'Anecdotal examples' in Figure 1.

2. Figures.

Although the figures are of high standard, in some cases they are too complicated and hard to understand.

Figure 4. Not uniform axis scales could be misleading as they can hide real differences among the species. I understand the motivation of the authors to compare the values of the co-directional and head-on groups, but since the bar plot also shows the significance of the differences, uniform axis scales would be more accurate to use.

Figure 5. In my opinion; the right panel of the figure is enough to demonstrate the results. In the same panel, the white bars demonstrating the magnitude of differences look unnecessary, they are more confusing rather than being helpful.

In Supplementary Tables 3 and 4 the titles are inverted: the 'Table S3' title belongs to Supplementary Table 4 and vice versa.

Point by point:

Reviewer #1 (Remarks to the Author):

This is an interesting and thought-provoking study addressing the role of transcription-replication collisions in bacterial evolution. The main point is that large number of ancestral co-directional genes have inverted to the head-on orientation. The latter head-on genes have higher dN/dS ratio and are more frequently under positive selection. The authors proposed that bacteria increase their evolution rate through gene inversion resulting in head-on transcription-replication collisions.

The key advance of this study is the calculation of the GC skew for all chromosomally encoded genes in multiple important clinical pathogens. This approach allowed the authors to nicely document the inversion record of every single gene in each species. They found that bacterial genomes are consistently gaining new head-on genes by inversion of co-directional genes. In fact, their data indicate that major percentage of current head-on genes were originally co-directional.

They then analyzed the rates of nonsynonymous (dN) and synonymous (dS) mutations among the core genes of each species, using at least 10 whole genomes for each. It appeared that dN/dS ratio is elevated in the head-on genes of each species relative to the corresponding co-directional genes. Furthermore, the fraction of genes with the dN/dS ratio above 1 (an indicator of the positive selection) is significantly higher in these head-on genes than in their co-directional counterparts. Finally, these head-on genes are enriched in common functions across species; specifically, they include important virulence and antibiotic resistance genes.

Overall, this is a very interesting study: the data are novel and they are by-and-large in agreement with the authors' conclusions. I have, however, few questions/comments that the authors may want to address.

First, DNA segments that undergo inversions are usually flanked by inverted DNA repeats. Do the authors see IR around co-directional genes that tend to undergo inversions?

We did not initially look for inverted repeats as part of the mechanistic basis for gene flipping. We have now conducted this analysis at the reviewer's request by blasting the up/downstream sequences surrounding putative flip regions. We compared these results to two negative controls: random intergenic regions blasted against each other, and open reading frame fragments from the same gene blasted against one another. We found a slight increase in the amount/length/degree of homology up/downstream of anticipated flip regions relative to the controls. However, as these data did not yield a strong signal, and are a bit outside the scope of the paper, we prefer not to include them (below).

Second, a thematically close study [Sankar et al (2016) The nature of mutations induced by replication–transcription collisions Nature 535: 178-181] found two mutational signatures for the head-on transcription replication collisions: duplications/deletions at the 3'-part of a head-on reporter and base substitutions in its promoter. Do the authors' analysis reveal these mutational signatures? Whatever the answer, this matter needs to be discussed.

The software we used for evolutionary analyses (TimeZone) does not identify indels, nor does it investigate promoter regions. As such, these analyses are not possible with our methods. Sankar et al. were only able to do this because they were doing a very different experiment - lab-based mutation assays with a known population founder sequence. The two studies are quite different in their nature.

Third, I cannot grasp how the authors' hypothesis can explain dramatic difference in terms of collisions between Gram-positive and Gram-negative bacteria. The authors are well aware of this difference stating that “Gram-positive species possess roughly 1 head-on gene for every 5 co-directional genes, whereas the ratio is as low as 1:1.4 for Gram-negative species”. In light of the authors' hypothesis this would mean that Gram-negative bacteria evolve much slower than Gram-positive bacteria, which is hardly the case. I'd like the authors to discuss this paradox.

We thank the authors for highlighting this point and we have updated the discussion to address these points. In the previous draft alluded to the possibility that the increased number of head-on genes in Gram negative bacteria might increase their overall mutation rate relative to Gram positive species. However, we realized that this was an overgeneralization and that mutation rates are in fact well established as being species-specific. Still, in all species tested, HO genes evolve faster than the same species' CD genes.

Regarding the apparent contradiction between the increased head-on conflict burden in Gram negative cells (imposed by the increased number of HO genes) and the apparent lack of major consequence, it appears that *E. coli* (and potentially other Gram-negative bacteria) are better suited to resolving (or tolerating) HO conflicts. Although the exact reasons for these differences are not understood, there are some potential explanations. For example, *E. coli* has more resolution mechanisms (3 accessory helicase, DinG, Rep and UvrD) than *B. subtilis* (two accessory helicase, DinG, PcrA). Refs: PMC2770101, PMC4466434.

Reviewer #2 (Remarks to the Author):

In the current manuscript, Christopher Merrih and Houra Merrih report a novel link between the chromosomal direction of bacterial genes and the adaptability of their host organisms.

It is a well-known fact that the gene composition of the leading and the lagging strand of the bacterial chromosomal DNA is different. On the lagging strand the direction of the replication and transcription machinery is opposite resulting frequent collision between the replisome and the RNA polymerases. The collision causes the collapse of the replication fork, subsequently leads to DNA break and thereby increases mutation rate in the so-called head-on oriented genes located on the lagging strand. In order to avoid the detrimental effect of machinery collision, the majority of bacterial genes are located on the leading strand; hence their transcription is co-directional with the replication. It is generally assumed that there is a negative selection pressure against gene inversion events which is responsible for the head-on orientation of genes. Accordingly, frequently-expressed and essential genes tend to be co-directional (Wen-Xin Zheng et al, 2015) Christopher Merrih and Houra Merrih aim to challenge the above-mentioned findings in their recent manuscript. The authors report that the head-on orientation of specific genes might be beneficial in certain cases by increasing the likelihood of the accumulation of adaptive mutations. They claim: The head-on orientation of certain genes causes increased mutation rate that helps the adaptation of cells. The head-on gene orientation is very frequent among virulence-related and antibiotic resistance-conferring genes.

The head-on gene orientation is a previously uncovered mechanism among bacteria that can generally contribute to the rise of antibiotic resistance.

I found the subject of the paper interesting, but the general conclusions are not supported by their results. The

presented results failed to convince me that the head-on gene orientation has positive effect and thereby it is adaptive. The results only demonstrate that the head-on gene orientation is not always detrimental.

1. The authors only showed that virulence-related genes are enriched among head-on oriented genes, but they did not demonstrate any bias toward head-on direction within the virulence-related gene group.

2. The results demonstrate that the resistance-conferring genes can tolerate the head-on orientation, but there is no obvious that head-on inversion of originally co-directional genes can be beneficial for pathogenic bacteria.

Regarding the above points 1 and 2: We have added new data to address these points. Please see our discussion below ("Major Comments" - Point 3, and within the revised manuscript new data in Tables 2 and 3)

In general, I find the theoretical argument unconvincing. Do the authors really believe that inversion events are favored by selection for the future benefit of increasing evolvability?

We address this more clearly in the text now: we are not suggesting that inversions occur in order to increase evolvability. Rather, our data suggest that inversions occur randomly, and that certain inversions are retained if the resulting allele gains a beneficial mutation. This appears to occur more frequently in genes that were originally co-directional then flipped to the head-on orientation. We attribute this trend to the higher mutation rate of head-on alleles.

In my view it is implausible, because any potential selective advantage would be very low, for two reasons. First, the number of genes with increased mutation rate by this mechanism is limited.

All genes can be inverted so there is no theoretical limit. Also 20-47% of genes in every bacterial genome are currently head-on, so the selective advantage conferred by the head-on orientation applies to (minimally) at least 100 genes to 2200 genes in each genome, depending upon the species.

Second, it has remained unclear to what extent head-on orientation increases local mutation rate.

The effect of the head-on orientation on mutation rates was well studied in our lab and elsewhere (our lab: Paul et al., 2013 and Million-Weaver et al., 2015. Other labs: Sancar et al. 2016). We previously used three independent reporter genes to show that each mutated faster when oriented head-on. Critically, this elevated mutation rate occurred when the genes were head-on AND transcribed (mutation rates are equal and low in the absence of transcription.) This robustly isolated the effects of head-on replication-transcription conflicts. Our results were independently confirmed by Sancar et. al Nature 2016, Figure 1D, "Base substitutions" in the coding sequence (co-directional in blue vs. head-on in red).

To our knowledge, the only group suggesting that head-on genes do not evolve at a faster rate did not use adequate controls. e.g. Schroeder et al. (PMC4783269) failed to see higher mutation rates in head-on genes using mutation accumulation lines. This is an entirely distinct experimental setup from ours and thus their results are not mutually exclusive with ours. Further, they compare one group of naturally co-oriented genes to a wholly distinct group of naturally head-on genes. These comparisons are not necessarily valid, clouding the implications of their negative data.

Even more worrying, that there is a simple alternative possibility that needs to be considered. Head-on orientation is overall disadvantageous (e.g. due to interference with transcription), and only genes with relaxed selection can reside in the lagging strand.

Please see point 2 below.) This is a very important general consideration, and we thank the reviewer for raising it. **We now provide new data addressing the mutability hypothesis (Table S3). We show that an equal percentage of head-on and co-directional genes are highly conserved core genes. As such, head-on genes are evidently *not* more mutable** in any species examined. Therefore, our proposed model is the best available explanation for our findings.

Overall I find that the manuscript in its present form does not reach the high standard of Nature Communication, as the conclusion is not well-supported by the experiments and analyses. My concerns and suggestions are found below.

Major comments:

1. In order to detect gene inversion events, the authors used GC skew calculation. They mention that this

method is 'well-established' and 'previous work shows that such inversions can be identified by local changes in the GC skew'. However, they did not describe why they chose this method for their analysis.

- a) What are the alternative methods to identify gene inversion events? b) Why is the GC skew analysis assumed to be the best method to identify gene inversions (if it is)?
c) What are the limitations or the shortcomings of the GC skew analysis?

Regarding comment 1 and its three sub-points, we have added significantly to our discussion about the method and its advantages in the introduction. In the Introduction section the authors mention that in many cases positive selection acts on head-on genes. They showed that head-on genes tend to have a high non-synonymous/synonymous mutation ratio ($dN/dS > 1$). However, high dN/dS ratio around one may also reflect relaxed negative selection. Consistent with this hypothesis, non-essential genes tend to be head-on oriented.

Addressing a theme raised by referee #2 in several places, we have added both a description of an additional important null hypothesis in the text as well as new data indicating that it is not correct. Specifically, we consider the possibility that head-on genes can simply handle more mutations, resulting in the observed elevation in dN/dS ratios. **Our new data shows that that head-on genes are not more mutable**: the same percentage of head-on and co-directional genes are highly conserved core genes in every species tested (Table S3). This conservation analysis is also a very good measure of essentiality, and suggests that previous analyses may be wrong – essential genes are not enriched in the co-directional orientation. (Reference for the benefit of this method ("Putting essentiality into context", Nature Reviews in Genetics, 2017, PMID: 29230014 <https://www.nature.com/articles/nrg.2017.141>)

Specifically regarding the elevated dN/dS ratio of head-on genes we observe in all species: it is true that both increased mutability and positive selection (two distinct forces) could potentially result in a general increase in the dN/dS ratio (which we see). However, we also observe a higher percent of head-on genes have a dN/dS value in excess of 1. This is a second and critical observation that helps distinguish between the action of these two forces as only positive selection can drive the dN/dS ratio over 1. Therefore, **our data showing that head-on genes have a higher frequency of $dN/dS > 1$ genes (Table 1) is powerful evidence that head-on genes are more frequently under strong positive selection. It also suggests that the generally elevated dN/dS ratio of head-on genes could also be due to positive selection.**

In summary, our data fail to support the null hypothesis (new data, Table S3) and also strongly support our conclusions that head-on gene mutations provide a benefit to the cell, both generally, and also *more frequently* than co-directional mutations (Table 1).

Genes under relaxed negative selection are expected to be less sensitive to the elevated mutation rate, therefore the head-on orientation can be preserved during evolution. In the Chapter 'Head-on genes are retained over evolutionary time scales' the authors have demonstrated the same: inverted genes have been in head-on orientation for a long time.

3. The authors refer to literature that 'local DNA inversions can spontaneously occur in all living cells' and they write in Introduction section that 'bacterial genomes are consistently gaining new head-on genes and operons via inversion'. However, the appearance of new head-on oriented genes is not clearly proved by the results and the authors have only quoted 'anecdotal examples' demonstrating 'important virulence and antibiotic resistance genes are head-on' (last chapter of Results section).

We initially provided strong evidence that both $HO > CD$ and $CD > HO$ inversions have occurred in nature vis a vis our whole closed genome comparisons in Figure 1C. These are just four examples (out of many) of existing alleles from natural isolates of Mtb for which inverted alleles can be found in other strains. **We have simplified/clarified the figure and also provide the full list of 239 Mtb genes for which we identified both head-on and co-directional alleles in various strains (again by whole genome analysis)**

which is a distinct and complementary method to the GC skew analysis). This includes numerous examples of naturally co-oriented alleles that have inverted to become head-on, thereby gaining a negative GC skew (Table S1).

The authors should demonstrate the ratio of co-direction to head-on conversion generally as well as within virulence-related genes.

We would like to thank the reviewer for this excellent suggestion. We did this analysis and now provide these data as a last figure (Tables 2 and 3). The data turned out to be entirely in keeping with our model, and really nail down our key conclusions – that head-on virulence genes evolve at a faster rate than co-directional virulence genes. The same is true for antibiotic resistance genes.

A higher ratio of head-on orientation of virulence-related genes in pathogenic bacteria would support the hypothesis that head-on orientation of specific genes is indeed preferable in some cases. Furthermore, one might also expect that the gene inversion of resistance-related genes should be more frequent in bacteria that encountered antibiotic exposure in the clinic than in those that did not. Without the above-mentioned demonstration, I am not convinced that abundant co-directional to head-on inversions due to positive selection pressure have been occurred recently.

This is a very nice idea but it would be almost impossible to delineate which naturally occurring bacterial strains were exposed to drugs from those that were not.

Regarding the use of the term “recently”, we are using this as a relative term. These inversions could have occurred anywhere from tens of years ago to millions of years ago. It is simply unclear from our data when these inversions occurred as the necessary molecular clock analyses are beyond the scope of this study. What is very clear from our work is that some genes have been retained for extensive time periods (and have a positive GC skew), while others have flipped at some point (making the GC skew negative). For these genes, the GC skew values have not yet risen to become fully positive as they eventually should.

4. One of the main findings of the manuscript is that the head-on orientation of genes increases the virulence and antibiotic resistance in bacteria. At the end of Introduction section, the authors write that ‘bacteria are universally increasing the mutability of their genome via inversions’ and it is a ‘commonly harnessed’ procedure among bacteria. However, I find that this conclusion is premature and exaggerated. Later on, the authors themselves restrained their previous statement in the Chapter ‘Head-on genes are enriched in common functions across species’: ‘This suggests that many virulence and antibiotic resistance genes may evolve at an accelerated rate, at least partially due to their head-on orientation.’ These claims are not directly supported by the data.

Please see above (#3).

5. The authors conclude that special functions are enriched among the head-on genes. They found only four cell-physiology functions which are enriched among the head-on genes including toxin and virulence genes. They claim that bacteria adapt faster by the higher mutation rate of these special genes. In my opinion; the authors should analyze the average mutation rate of virulence-related genes irrespective of their orientation. This analysis would be able to clarify, whether the higher mutation rate of virulence-related genes is a consequence of gene orientation or the gene function itself.

Minor comments:

1. Lack of references.

The authors refer to previous results many times without references in the Result section. It is not clear, whether or not these are the authors’ previous studies.

Updated.

Page 4. 'As previously shown in a variety of bacterial species, guanine nucleotides outnumber cytosine nucleotides on the leading strand of each arm of the chromosome.'

Page 9. '...we previously showed that head-on conflicts increase gene specific mutation rates and accelerate evolution in *B. subtilis* in nature.'

Page 12. 'Previous work demonstrated that the negative GC skew of inverted DNA fragments rises due to normal replication, just like any other DNA fragment.'

Page 6. 'Anecdotal examples' in Figure 1.

2. Figures.

Although the figures are of high standard, in some cases they are too complicated and hard to understand.

We have updated the figures and legends for clarity.

Figure 4. Not uniform axis scales could be misleading as they can hide real differences among the species. I understand the motivation of the authors to compare the values of the co-directional and head-on groups, but since the bar plot also shows the significance of the differences, uniform axis scales would be more accurate to use.

Figure 5. In my opinion; the right panel of the figure is enough to demonstrate the results. In the same panel, the white bars demonstrating the magnitude of differences look unnecessary, they are more confusing rather than being helpful.

Fixed.

In Supplementary Tables 3 and 4 the titles are inverted: the 'Table S3' title belongs to Supplementary Table 4 and vice versa.

Fixed.

REVIEWERS' COMMENTS:

Reviewer #1 (Remarks to the Author):

The revised manuscript is much improved by the addition of new data that address many of Rev. 2 comments and by revamping the Discussion section. The mechanisms leading to gene inversions, however, remain elusive, particularly given the authors' new data (included in the response to the reviewers) showing only a slight increase in the extent of homology in sequences flanking the flipped genes. One possible mechanism accounting for gene flipping could be a concept of "replication punctuation marks", which stipulates that replication forks appear to preferentially stall at gene promoters and terminators. As fork stalling can lead to DSB formation, these breaks at gene flanks could result in their inversions via microhomology-mediated end-joining (MMEJ). In this scenario, one should not expect significant sequence homology between the flanking regions of flipping genes. This model could potentially be extended to explain the difference in the fraction of HO genes between gram-negative and gram-positive bacteria. As the authors state, gram-negative bacteria are better suited to resolve transcription-replication collisions than their gram-positive counterparts. In other words, the occurrence of DNA breaks at gene flanks could be more profound in gram-positive bacteria, resulting in more frequent flipping subsequently picked-up by selection.

Reviewer #2 (Remarks to the Author):

I appreciate the new approaches in the re-written manuscript, including the discussion of alternative explanations beyond their own hypothesis and by not over-generalizing their findings. Moreover, addition of the section 'Gene function, but not essentiality, drives gene-strand bias' in the Discussion Section will be useful for the readers. The detailed explanation will solve a potential conflict between the result of previous investigations (head-on orientation is detrimental in general) and the current study (in special cases head-on orientation is adaptive). The manuscript is written in a clear and logical style, the corrected figures and tables are far easier to interpret and the results support the hypothesis.

Figure 1C is a lot easier to understand and the explanation referring to the figure is also well-detailed. In the main text the authors demonstrated that gene inversion is a recent evolutionary event. One of my main concerns was that the identified head-on oriented genes basically have a higher mutation rate and therefore they simply tolerate the detrimental head-on inversion. However, in a new analysis the authors demonstrate that the conservation rate of head-on genes is the same as the conservation rate of co-directional genes. It proves that head-on genes are not more prone to mutations (Table S3).

My other concern focused on the enrichment of resistance-conferring and virulence-related genes among head-on genes. The authors now analyzed the above mentioned gene groups individually. They found that these genes in head-on orientation have much higher mutation rates than their co-directional counterparts (Tables 2 and 3). Despite the high mutation rate, these genes retain their head-on orientation permanently which proves that inversion is beneficial.

In Supplementary Table 6 the authors mention that 'GC skew values for each gene are also presented', but the values are not visible. Please, check it and correct them.